# 5-Hydroxytryptophan (5-HTP): Natural Occurrence, Analysis, Biosynthesis, Biotechnology, Physiology and Toxicology

**DOI:** 10.3390/ijms22010181

**Published:** 2020-12-26

**Authors:** Massimo E. Maffei

**Affiliations:** Department of Life Sciences and Systems Biology, University of Turin, Via Quarello 15/a, 10135 Turin, Italy; massimo.maffei@unito.it; Tel.: +39-011-670-5967

**Keywords:** 5-hydroxytryptophan, natural sources, microbial production, biosynthetic pathways, physiological effects, animal, human

## Abstract

L-5-hydroxytryptophan (5-HTP) is both a drug and a natural component of some dietary supplements. 5-HTP is produced from tryptophan by tryptophan hydroxylase (TPH), which is present in two isoforms (TPH1 and TPH2). Decarboxylation of 5-HTP yields serotonin (5-hydroxytryptamine, 5-HT) that is further transformed to melatonin (*N*-acetyl-5-methoxytryptamine). 5-HTP plays a major role both in neurologic and metabolic diseases and its synthesis from tryptophan represents the limiting step in serotonin and melatonin biosynthesis. In this review, after an look at the main natural sources of 5-HTP, the chemical analysis and synthesis, biosynthesis and microbial production of 5-HTP by molecular engineering will be described. The physiological effects of 5-HTP are discussed in both animal studies and human clinical trials. The physiological role of 5-HTP in the treatment of depression, anxiety, panic, sleep disorders, obesity, myoclonus and serotonin syndrome are also discussed. 5-HTP toxicity and the occurrence of toxic impurities present in tryptophan and 5-HTP preparations are also discussed.

## 1. Introduction

L-5-hydroxytryptophan (5-HTP) is produced from tryptophan by tryptophan hydroxylase (TPH) and its decarboxylation yields serotonin (5-hydroxytryptamine, 5-HT), a monoamine neurotransmitter involved in the modulation of mood, cognition, reward, learning, memory, sleep and numerous other physiological processes [1]. 5-HT is further transformed to melatonin (*N*-acetyl-5-methoxytryptamine), the hormone primarily released by the pineal gland that regulates the sleep–wake cycle [2,3]. Therefore, the biosynthesis of 5-HTP is important and necessary for the production of key molecules such as 5-HT and melatonin. 5-HTP also plays a major role both in neurologic and metabolic diseases and its synthesis from tryptophan represents the limiting step in 5-HT and melatonin biosynthesis [4]. The occurrence of 5-HTP is not limited to animals or humans and the molecule is produced by lower and higher plants, mushrooms and microbes (see below).

5-HTP is both a drug and a natural component of some dietary supplements and occurrence in both synthetic tryptophan and 5-HTP of toxic impurities has caused eosinophilia myalgia syndrome cases [5,6,7,8]; therefore, accurate chemical analyses and characterization have been developed. Recently, microbial engineering allowed the production of 5-HTP from alternative biosynthetic routes, opening the interesting possibility of better controlling the presence of contaminants.

This review describes the natural sources of 5-HTP, which represent the raw material from which 5-HTP may be extracted and purified for its use in drugs and dietary supplements, as well as alternative microbial synthesis that is considered a stable and sustainable way to provide a constant supply of the molecule. Chemical synthesis and chemical analysis of 5-HTP are discussed, along with the major biochemical pathways involved in 5-HTP biosynthesis and transformation, with particular reference to tryptophan hydroxylase. The physiological role of 5-HTP is discussed by considering both animal studies and human clinical trials. The toxicology of 5-HTP and the potential occurrence of toxic impurities are also discussed.

## 2. Databases, Exclusion and Inclusion Criteria

The strategy that was implemented to carry out this review was based on a deep search in the databases Web of Science (1985–2020) and PubMed (1940–2020) by considering, as the main entry, the term 5-hydroxytryptophan. The total number of Web of Science Core Collection papers was 1857, whereas the total number in PubMed was 5431 papers. There were 816 selected papers in the first search (see Appendix A) and the exclusion criteria were the impossibility to obtain a full text and the lack of specificity with respect to the selected areas of the review. Out of the 816 papers, 195 were used for this review because of the historical, breakthrough and innovative content.

## 3. Natural Sources of 5-HTP

Plants are a rich source of 5-HTP and *Griffonia simplicifolia* Baill. (Caesalpinaceae) (also known as the alternate incorrect name *Bandeiraea simplicifolia*) seeds are the most used for the extraction and the commercial production of 5-HTP. Preparations of *G. simplicifolia* containing high concentrations of 5-HTP are used to treat serotonin-related disorders, including motion sickness [9] and to increase the feeling of satiety associated with a significant reduction of food intake and body weight with decreasing body mass index (BMI) [10,11]. The extract also shows anxiolytic-like effects [12]. The authentication of *G. simplicifolia* seeds is based on methods that are often laborious, time-consuming and sensitive to interference from co-occurring materials. Rapid and simple estimation procedures have been developed for the identification and quantification of 5-HTP in *G. simplicifolia* extracts [13,14]. Recently, the combination of chemical analysis (with the characterization of 5-HTP along with the β-carboline alkaloid derivatives) and molecular DNA fingerprinting (DNA restriction fragment length polymorphism—PCR-RFLP—analysis which has been performed on the plant internal transcribed spacer—ITS) allowed the unequivocal identification of commercial *G. simplicifolia* seeds [15]. 

Both 5-HTP and 5-hydroxytryptophan hydrate have been identified in the root allelochemical exudates from the aggressive weed couch grass, *Elytrigia* (*Agropyron*) *repens* [16], whereas quack grass (*Agropyron repens* L. Beauv.) accumulates throughout the plant high levels of 5-HTP as glucosides attached in β-linkages to the 5-O-indolyl moiety [17]. St. John’s wort (*Hypericum perforatum* cv. Anthos) stem explants were found to produce significant amounts of 5-HTP when plantlets were regenerated from thidiazuron-induced tissue grown on a basal culture medium for 2 months [18], whereas 5-HTP was detected during the alcoholic fermentation of some grape cultivars [19]. Food processing may alter the 5-HTP content. The cooking (boiling, steaming, and microwaving) of the cauliflower genotypes Forata (white inflorescence), Verde di Macerata (green inflorescence), Cheddar F1 (yellow inflorescence), and Graffiti (purple coloration) was found to increase tryptophan levels and to reduce the content of 5-HTP [20]. 

Interestingly, other organisms are potential sources of 5-HTP. The intertidal sponge *Hymeniacidon heliophila*, which survives under intense sunlight, contains 5-HTP as a major constituent [21], whereas methanolic extracts of the mushrooms *Boletus edulis*, *Suillus luteus*, and *Pleurotus ostreatus* contain, among other indole compounds, fairly good amounts of 5-HTP [22]. The fruiting bodies of the mushroom *Cantharellus cibarius* (the chanterelle) and the mycelium of this species cultured in vitro contain eight indole compounds, including 5-HTP [23], while the concentration of 5-HTP was higher in the stipes of five species of the fungal genus *Panaeolus* (*P. ater*, *P. rickenii*, *P. papilionaceous*, *P. sphinctrinus*, and *P. subbalteatus*) [24].

## 4. Qualitative and Quantitative Analysis of 5-HTP

The chemical analysis of 5-HTP is mainly performed by using high-performance liquid chromatography (HPLC) coupled to different detectors, including diode array (DAD), fluorescence (FD), and mass spectrometry (MS) detectors. The separation is usually performed by reverse phase C-18 column chromatography [25]. The binary solvent system is H_2_O acidified with 0.1% *v*/*v* formic acid (solvent A) and acetonitrile acidified with 0.1% *v*/*v* formic acid (solvent B). The chromatographic profiles are usually registered at 230 and 270 nm. Quantitative analysis is usually performed by positive ion mode, often using a selective ion monitoring (SIM) method with different ionization interfaces, the most used being electrospray ionization (EI) [15]. This method, along with the use of other detectors, has been used to detect and quantify 5-HTP in plant extracts [2,14,15,26], rat serum [25], mice whole brain tissue [27], human plasma [28], human urine [29], and in the central nervous system [30,31].

Besides HPLC, other chromatographic techniques have been used to detect and quantify 5-HPT. Capillary electrophoresis (CE) has been used with different detectors to determine 5-HTP in samples of commercial dietary supplements [32], in human platelet-rich plasma [33], and immortalized rat raphe nuclei neurons [34]. A rapid method for the separation of 5-HTP was developed by using micellar electrokinetic chromatography with diode array detection [35]. The identification and quantification of 5-HTP was also achieved by the combination of solid-phase extraction pretreatment and gas chromatography–mass spectrometry based on a modified method of derivatization by silanization [36].

An NMR-based approach showed that 5-HTP results in characteristic chemical shift correlations suited for its identification and quantification [37], whereas the quantitative analysis of 5-HTP of carcinoid tumors was assayed using gold nanoparticles as the assisted matrix in surface-assisted laser desorption/ionization time-of-flight mass spectrometry [38]. A green fluorescence transient 5-HTP is obtained by multiphoton near infrared excitation and this technique enables the detection of 5-HTP with extremely high sensitivity. The potential application of such a method is in the imaging of biological systems and the investigation of protein dynamics [39]. Both cyclic voltammetric and UV–visible spectroscopic methods have been demonstrated to show linear responses over a wide concentration range of 5-HTP, with low limits of detection also in dietary supplements [40]. 

Electrochemical sensors have also attracted much attention for the detection and quantification of 5-HTP. Their high sensitivity and miniaturization are expected to overcome the shortcomings of high-cost, time-consuming, and complicated operations linked to chromatographic methods. Various types of surface-modified electrodes have been developed to determine 5-HTP such as a carbon nanosheet-modified electrode [41], Ru^II^terpyridine-doped composite electrode [42], gold-modified pencil graphite electrode [43], graphene-chitosan molecularly imprinted film modified on the surface of a glassy carbon electrode [44], sensors based on electro-polymerization to obtain a poly-(melamine)/poly-(o-aminophenol) co-polymeric film [45], nano-palladium decorated multi-walled carbon nanotubes [46], citrate-capped gold nanoparticles [47], a pyrolytic graphite electrode with the surface covered with a thin film of a nano-mixture of graphite/diamond [48], and by electrochemical microfluidic separation and sensing [49].

## 5. Chemical Synthesis of 5-HTP

The synthesis of 5-HTP by the condensation of 5-benzyloxygramine with diethyl formaminomalonate, followed by saponification, decarboxylation, and hydrogenolysis was described in 1951 [50] and 1954 [51] and was an application of gramine synthesis, developed by Snyder and Smith ten years before [52]. A few years later, another application of gramine synthesis was reported [53]. In the same year, Frangatos and Chubb [54] reported an application of the convenient tryptophan synthesis developed ten years before [55] by eliminating the difficult and tedious preparation of 5-benzyloxyindole. The *p*-benzyloxyphenylhydrazone of γ,γ-dicarbethoxy-γ-acetamido-butyraldehyde (Figure 1, **I**) was prepared and cyclized, without isolation, to form ethyl β-(5-benzyloxyindol-3-)-α-carbethoxy-α-acetamidopropioilate (Figure 1, **II**). Saponification and partial decarboxylation of **II**, followed by hydrolysis of the acetamido group, gave 5-benzyloxytryptophan (Figure 1, **III**). 5-HTP was obtained by hydrogenolysis of **III** (Figure 1). However, this synthetic method suffers from the difficulty involved in the regioselective hydroxylation of tryptophan.

## 6. Biosynthesis of 5-HTP and Inhibition of Tryptophan Hydroxylase (TPH)

The biosynthesis of 5-HTP starts with the essential amino acid tryptophan, which is metabolized to 5-HTP by TPH in an initial, rate-limiting step in the biosynthesis of serotonin after the decarboxylation catalyzed by aromatic amino acid decarboxylase (AADC). TPH is a monooxygenase that belongs to the family of aromatic amino acid hydroxylases; it incorporates one atom of oxygen from molecular oxygen into the substrate and reduces the other atom to water. The two electrons required for the reduction of the second atom to water are supplied by tetrahydrobiopterin (BH_4_), which acts as a substrate rather than a tightly bound cofactor, binding and dissociating each turnover [56]. The irreversible activation of O_2_ is the initial step in this mechanism and utilizes two electrons from BH_4_ to form a high-valent Fe(IV)O (ferryl) hydroxylating intermediate and 4a-hydroxypterin (4a-HOPH_3_). The Fe(IV)O intermediate subsequently reacts with the side chain of the aromatic amino acid through electrophilic aromatic substitution [57,58]. The binding of both the amino acid and BH_4_ results in a change in the coordination of the iron from six-coordinate to five-coordinate, presumably opening a coordination site for oxygen. The hydroxylating intermediate Fe(IV)O in TPH has been confirmed by rapid freeze-quench ^57^Fe Mössbauer spectroscopy [57]. During L-tryptophan hydroxylation, BH_4_ is oxidized to pterin-4α-carbinolamine (BH_3_OH) and regenerated through the function of pterin-4α-carbinolamine dehydratase (PCD) and dihydropteridine reductase (DHPR) [59,60]. Figure 2 shows the involvement of BH_4_ and the Fe intermediate in the reaction catalyzed by TPH.

In humans, BH_4_ is synthesized from guanosine triphosphate (GTP) via a three-step pathway, containing GTP cyclohydrolase I (GCHI), 6-pyruvate-tetrahydropterin synthase (PTPS), and sepiapterin reductase (SPR) [61], as shown in Figure 3.

TPH activity is assayed by a continuous spectrophotometric method that exploits the different spectral properties of tryptophan and 5-HTP [62]. The sensitivity of the essay allows the use of relatively low enzyme concentrations and can be used to determine the steady-state kinetic parameters for each of the enzyme substrates [62].

TPH is composed of three functional domains, a regulatory *N*-terminal domain, a catalytic domain, and a *C*-terminal oligomerization domain [63]. TPH requires ferrous iron for activity. The activity of the enzyme is affected by phosphorylation and the requirement for Ca^2+^ suggests a need for a calcium-dependent kinase [56]. In the rat pineal gland, norepinephrine stimulates TPH synthesis and activity and cAMP through the activation of cAMP-dependent protein kinase A (PKA), phosphorylates the transcription factor cAMP response element binding protein (CREB), which starts the enzyme synthesis, and the incorporation of at least 1 mol of phosphate/mol of tetramer of native TPH is required for maximal activation [64].

In vertebrates, there are two molecular forms of TPH: TPH1 is responsible for serotonin synthesis in peripheral tissues and is mainly expressed in the enterochromaffin cells of the gut and in the pineal gland. TPH1-expressing cells of the gastrointestinal (GI) tract are responsible for blood 5-HT synthesis; 5-HTP then enters the circulation packed in dense granules of thrombocytes where it mediates its hormonal actions upon platelet release at the site of activation [65]. The second form, TPH2, is expressed in peripheral myenteric neurons in the gut and in the neurons of raphe nuclei in the brain stem but not in peripheral organs (lung, heart, kidney, or liver) [4]. Cells expressing TPH2 have rates of 5-HT synthesis which are affected by changes in tryptophan availability [66]. The presence of these two TPH forms justifies the duality of the 5-HT system, with two independently generated pools of 5-HT—one in the brain and another in the blood [65]. Antibodies that distinguish between the isoforms of TPH have been developed [67]. Figure 4 shows the central role of 5-HTP in the biochemical pathway of 5-HT.

Because TPH inhibitors may provide novel treatments for various gastrointestinal disorders associated with dysregulation of the gastrointestinal serotonergic system, such as chemotherapy-induced emesis and irritable bowel syndrome, both academia and the pharmaceutical industry have worked on the search for specific TPH inhibitors. Naturally occurring unspecific inhibitors of TPH (and indoleamine metabolism) have been reported, including catecholamines, the food-derived carcinogenic heterocyclic amines 3-amino-1,4-dimethyl-5H-pyrido[4,3-b]indole (Trp-P-1), and 3-amino-1-methyl-5H-pyrido[4,3-b]indole (Trp-P-2), as well as the dopamine-derived tetrahydroisoquinolines, such as salsolinol and tetrahydropapaverine [68,69,70]. While the inhibition of TPH by salsolinols was found to be non-competitive with the substrate L-tryptophan [68], TPH was un-competitively inhibited by tetrahydropapaverine with the substrate L-tryptophan, and non-competitively inhibited with the cofactor DL-6-methyl-5,6,7,8-tetrahydropteridin in P-815 cells [69]. In the same cell system, the inhibition of TPH by Trp-P-2 was found to be competitive with the substrate L-tryptophan and non-competitive with the cofactor DL-6-methyl-5,6,7,8-tetrahydropteridin [70]. *p*-Ethynylphenylalanine (*p*-EPA) is a more potent TPH inhibitor; *p*-EPA injection induced a significant and gradual decrease in extracellular 5-HTP in the rat hippocampus, striatum, and frontal cortex. Moreover, *p*-EPA could also irreversibly interfere with the synthesis of TPH [71]. Selective inhibitors of TPH, such as LP-533401 [(2S)-2-amino-3-(4-(2-amino-6-(2,2,2-trifluoro-1-(3′-fluorobiphenyl-4-yl)ethoxy)pyrimidin-4-yl)phenyl)propanoic acid] and LP-615819 [(2S)-ethyl 2-amino-3-(4-(2-amino-6-(2,2,2-trifluoro-1-(3′-fluorobiphenyl-4-yl)ethoxy)pyrimidin-4-yl)phenyl)propanoic acid], were found to competitively bind to the tryptophan pocket of both TPH isoforms and to improve metabolic parameters, thus providing novel treatments for various gastrointestinal disorders associated with dysregulation of the gastrointestinal serotonergic system [72]. The kinetic analysis with these inhibitors showed that they are all competitive versus L-tryptophan but predominantly uncompetitive versus pterin [73]. Two other inhibitors of TPH are telotristat ethyl (the free base form of a hippurate salt called telotristat etiprate) and its active metabolite telotristat. In vitro, the inhibitory potency of telotristat was found to be 29-fold higher than that of its prodrug, with inhibition of TPH resulting in a reduced production of peripheral 5-HT [74].

A new series of acyl guanidines displaying potent TPH1 inhibition have been reported [75]; these molecules have been chemically and pharmacokinetically optimized and successfully tested in vivo [76]. 1-O-Galloylpedunculagin was screened as a drug-like compound from the traditional Chinese medicine (TCM) database for inhibitor activity on TPH. The molecule specifically inhibited TPH1 but was ineffective on TPH2, and the inhibitory action displayed characteristics of competitive inhibition [77]. The effects of nifedipine, an L-type calcium channel blocker, in noradrenergic-stimulated cultured rat pineal glands, showed that TPH activity was the main step inhibited by the molecule, demonstrating that the calcium influx through L-type high-voltage-activated calcium channels is essential for the full activation of the enzyme [78]. Figure 5 shows the chemical formulae of the TPH inhibitors cited above.

## 7. Metabolic Engineering and Heterologous Production of 5-HTP

As discussed above, extraction from the seeds of the African plant *Griffonia simplicifolia* is the typical approach for 5-HTP commercial production, because chemical synthesis is not economically feasible on a large scale. However, the material supply is seasonally and regionally dependent, which limits the output of 5-HTP. A promising alternative is the metabolic engineering of microorganisms. Among the many advantages, microorganisms grow quickly and their genetic engineering has been demonstrated to be a productive way to obtain important chemicals, such as aromatic compounds [79], fatty acids [80], carotenoids [81], flavors and fragrances [82], and many other natural products [83]. 

Although the microbial synthesis of 5-HTP has been achieved in different microorganisms, with particular reference to *Escherichia coli*, the first evidence of a microbial hydroxylation of tryptophan was found in *Chromobacterium violaceum*, about 70 years ago [84]. The function of tryptophan hydroxylation in this organism was considered important to provide the precursor for the characteristic blue pigment produced by this species, violacein, that is synthesized by a series of tryptophan metabolisms. Further characterization showed that the TPH from *C. violaceum* has, at least superficially, the characteristics of the TPH found in various mammalian tissues [85]. L-phenylalanine 4-hydroxylase from *C. violaceum* could convert L-tryptophan to 5-HTP and L-phenylalanine to L-tyrosine; however, the activity for L-tryptophan is extremely low compared to L-phenylalanine activity levels. The L-tryptophan hydroxylation activity of *C. violaceum* L-phenylalanine 4-hydroxylase (CviPAH) was enhanced using information on its crystal structures by introducing a saturation mutagenesis towards L101 and W180 in *C. violaceum* phenylalanine 4-hydroxylase (PAH). Mutant libraries from the *L101* and *W180* produced several positive mutants, with *L101Y* and *W180F* showing the highest TPH activity, whereas the double mutant (*L101Y-W180F*) displayed higher TPH activity when compared with the wild type and the individual *W180F* and *L101Y* mutants [86]. Interestingly, pterin is still required as a cofactor for enzyme activity by the *CviPAH-L101Y-W180F* triple mutant, which is similar to the requirements of other types of aromatic amino acid hydroxylases [86]. A novel cofactor regeneration process to achieve enhanced synthesis of 5-HTP by using CviPAH was obtained by screening and investigating several key enzymes, including dihydropteridine reductase from *E. coli*, glucose dehydrogenase from *Bacillus subtilis*, and pterin-4α-carbinolamine dehydratase from *Pseudomonas syringae*. Genes encoding these three enzymes were overexpressed in an *E. coli* tryptophanase-deficient host, resulting in the synthesis of ten-fold (0.74 mM) 5-HTP in the presence of 0.1 mM pterin with respect to the absence of the regeneration of pterin [59].

The engineering of *E. coli* successfully improved the production of 5-HTP through bioprospecting and protein engineering approaches. An important achievement was the discovery that bacterial PAHs may utilize tetrahydromonapterin (MH_4_, Figure 6) instead of BH_4_ as the native pterin coenzyme; this resulted in a great advantage because BH_4_ does not naturally occur in most bacteria [87]. 

Based on the potential ability of PAH to hydroxylate tryptophan, the development of PAH mutants highly active in converting tryptophan to 5-HTP allowed the establishment of an efficient 5-HTP production platform via further metabolic engineering efforts [88]. Whole-cell bioconversion allowed the high-level production of 5-HTP (1.1−1.2 g/L) from tryptophan in shake flasks, also allowing de novo 5-HTP biosynthesis from glucose [88] (Figure 7).

In a similar strategy, the tryptophan pathway was extended by using an engineered PAH from *Cupriavidus taiwanensis* (CtAAAH) and the production of 5-HTP was achieved by an endogenous cofactor with an artificial regeneration system [89].

In the search of a direct alternative precursor for the biosynthesis of 5-HTP, a novel salicylate 5-hydroxylase was used to convert the non-natural substrate anthranilate to 5-hydroxyanthranilate (5-HI) produced from glucose. To assess whether 5-HI may function as a precursor of 5-HTP, a medium copy number pCStrpDCBA plasmid was constructed in the *E. coli* strain BW2. Knockouts of *tnaA* (that prevents the products tryptophan and 5-HTP from degrading) and *trpE* (that blocks the native synthesis of anthranilic acid) harboring pCS-trpDCBA were used for the in vivo assay. *trpDCBA* was also cloned into a low copy number plasmid, yielding pSA-trpDCBA. The in vivo assay of the strain BW2 harboring pSA-trpDCBA accumulated the intermediate 5-hydroxyindole in the cultures, indicating that the reaction catalyzed by TrpB was a rate-limiting step. The de novo production of 5-HTP was then established by combining the full pathway and by adopting a two-stage strategy (see Figure 8) [90]. 

By engineering *E. coli* with heterologous TPH by using a truncated form of human TrpH2 with an E2K mutation for improved protein abundance, 5-HTP conversion from tryptophan was improved by protein engineering TPH [91]. Recently, in order to increase 5-HTP yield and stability of TPH, the tryptophan biosynthetic pathway was integrated into the *E. coli* genome. A 24.8% improvement compared to the original strain was obtained by manipulating the replication origin of the hydroxylation plasmid and by substitution of the promoter of *aroH^fbr^* gene encoding 3-deoxy-7-phosphoheptulonate synthase (that catalyzes the first step of tryptophan biosynthesis). The resulted recombinant strain TRPmut/pSCHTP-LMT was able to produce 1.61 g/L 5-HTP in shake flasks, compared to 0.160 g/L of previous metabolic engineering productions [92].

A successful production of 5-HTP in microbial systems has been obtained in the yeast *Saccharomyces cerevisiae* BY4741 strain via LiAc-mediated yeast transformation. The strategy was based on the heterologous expression of either a prokaryotic PAH or eukaryotic tryptophan 3/5-hydroxylase, together with enhanced synthesis of the two cofactors MH_4_ or BH_4_ [93]. Interestingly, a native *S. cerevisiae* gene, *DFR1*, which encodes dihydrofolate reductase to catalyze tetrahydrofolate, played a pivotal role in 5-HTP synthesis by regenerating MH_4_ [93]. Figure 9 summarizes the heterologous 5-HTP production.

## 8. Physiological Effects of 5-HTP

### 8.1. Animal Studies

Early experiments with 5-HTP in the 1950s have shown that this molecule is also capable of inhibiting gastric hydrochloric acid secretion [94] and that when administered to animals it increased peristaltic activity [95] and was rapidly taken up by most tissues [96], producing in dogs, cats, rabbits, rats, and mice somatic, autonomic, and behavioral effects which grossly resembled those of lysergic acid diethylamide [97]. 

Most of the animal experimentation on the effects of 5-HTP has been conducted on mice and rats.

In mice, injection of 5-HTP produces a characteristic head twitch due to a central action of 5-HT formed by decarboxylation of 5-HTP [98] and provokes characteristic behaviors, such as tremors, that become more frequent when doses are increased [99]. The accumulation of 5-HT in the mouse liver but not in the brain is causally related to the hypoglycemia induced by 5-HTP [100,101]. 5-HTP also suppresses inflammation and arthritis through decreasing the production of pro-inflammatory mediators [102] and the antihistaminic drugs chloropyramine and, more strongly, chlorpheniramine potentiates the action of 5-HTP by inhibiting serotonin uptake [103]. Interestingly, dietary phenylalanine and 5-HTP were found to be mutually antagonistic in modulating mice audiogenic seizure susceptibility [104]. 

In rats, at least some of the behavioral effects of 5-HTP are due to increased levels or turnover of 5-HTP in peripheral serotonergic neuronal systems [105] and 5-HTP neurotoxicity caused by brain–blood barrier breakdown, edema formation, and NO production was found to be instrumental in causing adverse mental and behavioral abnormalities [106]. For instance, the wet dog shake behavior induced by 5-HTP was dose dependent and was mediated by the activation of 5-HT2 receptors [107], whereas in iproniazid-pretreated rats, a special form of stereotyped movements of the head and forelegs and a gnawing behavior were observed after intraperitoneal administration of 5-HTP [108]. 5-HTP produced a 6–11-fold increase in rat plasma prolactin [109,110]. Moreover, endogenous opioids also increase the serum level of prolactin induced by 5-HTP; therefore, different serotonergic neurotransmitter circuits might be capable of modulating the release of corticosterone and prolactin [111]. 5-HTP was also able to induce depression in rats working on an operant schedule for milk reinforcement, and this effect was mediated by serotonergic mechanisms involving 5-HT2 receptors [112]. Moreover, administration of 5-HTP prompted a synergistic increase in the synthesis and release of 5-HT by combining 5-HT uptake inhibition with the blockade of 5-HT1A autoreceptors [113]. Another interesting effect of 5-HTP in rats is the reduction in meal size and a slowing of eating, exerting an effect on the patterns of feeding [114].

Myoclonic twitches, jerks, or seizures are usually caused by brief lapses of contraction (negative myoclonus) or sudden muscle contractions (positive myoclonus). In guinea pigs, when 5-HTP was given 6 h after 0.5 mg progesterone in estradiol benzoate-primed males, myoclonus was enhanced, whereas progesterone reversed the facilitative effect of estradiol benzoate on 5-HTP-induced myoclonus in females [115]. It was also found that 5-HTP-induced myoclonus was influenced by the 5-HT1/2 receptor systems and that the absence of a significant change with a receptor antagonist implied that myoclonus was not related to diffuse activation of central serotonergic mechanisms [116]. In the guinea pig colon, 5-HTP facilitates the luminal 5-HT release from enterochromaffin cells, with no involvement of neuronal mechanisms and a non-neuronal cholinergic system [117].

In the terrestrial snail *Helix lucorum*, injection of 5-HTP alone did not restore the protein kinase M zeta (ZIP)- or the protein synthesis blocker anisomycin-impaired context memory, while the combination of 5-HTP and the reactivation of memory effectively reinstated the context memory [118].

In rabbits, 5-HTP injected intracisternally at a dose of 1.5–3 mg produced a fall in temperature often followed by a rise beyond the pre-injection level, and the anterior hypothalamus was supposed to be the site where 5-HTP acted [119]. To antagonize rabbit 5-HTP-induced hyperthermia, 5-HT receptor blockade is required and antagonism of *p*-methoxyamphetamine-induced hyperthermia is primarily a result of influence on the 5-HT system [120]. 

In cats, intrahypothalamic injection of 5-HTP allowed its detection in many fibers surrounding the injection site [121], whereas lysergic acid diethylamide was found to mimic the actions of 5-HTP by facilitating the stretch reflex and exciting extensor gamma motoneurons in the animal spine [122].

In dogs, the administration of 5-HTP increased both blood and tissue serotonin with a maximum effect evident in about one hour [123] and increased the propulsive activity, the contractile force, and the motility index [124]. In monoamine oxidase (MAO)-inhibited dogs, 5-HTP caused hypotension with variable effects on heart rate and the cerebral decarboxylation and formation of 5-HT was found to be responsible for this effect [125]. 

In sheep, the administration of 5-HTP substantially increases serum melatonin through a marked increase in pineal 5-HT and its metabolites, including *N*-acetylserotonin [126], whereas during the natural light period, a promotion of melatonin synthesis in the pineal gland and intestinal tract was found in rumen-protected 5-HTP [127]. In fetal lambs in late gestation, systemic infusion of 5-HTP during normoxia greatly increases the incidence of fetal breathing movements [128]. 

Finally, in the Holstein dairy cow liver, 5-HTP infusions stimulated an autocrine–paracrine adaptation to lactation [129].

Figure 10 summarizes some effects of 5-HTP on animals.

### 8.2. Effects of 5-HTP on Humans

#### 8.2.1. Serotonin Syndrome

While on the one side, the lack of serotonin is responsible for several diseases, including depression, its excess may be problematic. Medication-induced serotonergic hyperactivity causes serotonin syndrome, which results from antidepressant medications and is characterized by the triad of altered mental status, autonomic dysfunction, and neuromuscular abnormalities. Serotonin syndrome may lead to misdiagnosis, in a similar way as for neuroleptic malignant syndrome. Serotonin syndrome may eventually result in death; however, supportive care alone is sufficient to recover completely for most of patients. Excessive 5-HTP stimulation is the main pathophysiologic mechanism involved and the use of a serotonin antagonist supports this finding [130]. Serotonin syndrome can trigger other clinical conditions; therefore, in order to detect the syndrome and prevent rapid clinical deterioration, it is important to better understand the molecular context of this condition [131].

#### 8.2.2. Effect of 5-HTP on Depression, Anxiety, Dystonia, and Panic Disorders

The amount of endogenous 5-HTP available for serotonin synthesis depends on the availability of tryptophan and the activity of various enzymes, especially TPH, indoleamine 2,3-dioxygenase, and tryptophan 2,3-dioxygenase (TDO) [132]. In depressed patients, tryptophan, serotonin, kynurenine, and their metabolite levels remain unclear. Serotonin is involved in depressive pathophysiology and evidence indicates that the transport of 5-HTP across the blood–brain barrier is compromised in major depression [133], as found in childhood with major depression [134]. Early studies assessed the important role of tryptophan and 5-HTP as an antidepressant [135,136], and relatively few adverse effects are associated with its use in the treatment of depressed patients [137]. Among the side effects, the administration of 5-HTP may cause dose-dependent gastrointestinal problems, whereas the combination of 5-HTP with a peripheral decarboxylase inhibitor may cause psychopathological side effects, like acute anxiety [138]. Additionally, vomiting and nausea have been reported when 5-HTP was used at doses above 100 mg [139].

The antidepressant activity appears to be linked to the activation rather than suppression of monoaminergic activity; therefore, the decreased monoamine metabolism found in some types of depression is likely to depend on a primary metabolic deficit rather than receptor hypersensitivity [140]. Decreased serotonergic activity may be present in both depression and mania [141] and the therapeutic effect of 5-HTP has been correlated with an increase in serotonin at central serotonin receptors [142]. 

Evidence also supports the combined used of 5-HTP with other drugs. In 30 hospitalized patients affected by endogenous depression, the antidepressant action of the combination of nialamide and 5-HTP has been evaluated and compared with a control group which only received nialamide (along with placebo). The combination of nialamide and 5-HTP prompted a fuller recovery in treated patients with respect to those who were treated with nialamide alone [143]. L-deprenil (an irreversible selective MAO-B inhibitor) was used in an open trial study with patients with unipolar and bipolar depression receiving 5-HTP and benzerazide. The combination of L-deprenil and 5-HTP showed a significantly greater clinical improvement in treated patients with respect to placebo patients but not in patients treated with 5-HTP alone [144]. In women experiencing selective serotonin reuptake inhibitor (SSRI)- or serotonin–norepinephrine reuptake inhibitor (SNRIs)-resistant depression, the combination treatment with creatine and 5-HTP proved to be an effective augmentation strategy [145]. 

5-HT was found to mediate anxiety [146] and showed a moderate reduction of the symptomatology on the 90-item symptoms checklist (SCL-90) and the state scale of the Spielberger State–Trait Anxiety Inventory, suggesting that brain serotonergic pathways are involved in the pathogenesis of anxiety disorders, particularly in agoraphobia and panic disorders [147]. 

The serotonergic system of the central nervous system might play some role in the pathogenesis of dystonia in hereditary progressive dystonia [148], although results are at odds with the hypothesis that there is a supersensitivity of 5-HT2 receptors in panic disorder [149]. Nevertheless, in panic disorder patients, 5-HTP significantly reduced the reaction to the panic challenge, regarding number of panic attacks, panic symptom score, and subjective anxiety, when compared to placebo [150]. Furthermore, an increased availability of 5-HT may have a gender-dependent protective effect in cholecystokinin-tetrapeptide-induced panic [151].

#### 8.2.3. Effect of 5-HTP on Sleep Disorders

In normal subjects treated with 5-HTP, rapid eye movement (REM) sleep increased from 5 to 53% of placebo baseline [152] and the effects on sleep were associated with different doses of 5-HTP and the diverse possibilities of metabolic transformation of the precursor [153]. In schizophrenic boys, the administration of 5-HTP was associated with an increase in REM sleep and eye movements [154]; however, in mongoloid infants who received oral 5-HTP for periods extending from 12 to 36 months, 5-HTP failed to induce any long-term differences in the eye movement frequencies. In fact, the drug has a short-term effect lasting up to 8 days and an increase in muscle tone and an improvement of motor behavior were the only long-lasting results [155]. In a group of children with sleep terrors, treatment with 5-HTP was able to modulate the arousal level and to induce a long-term improvement of sleep terrors [156]. Interestingly, it was found that a serotonergic abnormality is involved in affective disorders [157] and direct modulation of the serotonergic system with 5-HTP was useful for the treatment of psychological suffering associated with unreciprocated romantic love [158].

#### 8.2.4. Effects of 5-HTP on Migraine, Ataxia, Fibromyalgia, Alzheimer’s, and Parkinson’s Disease

5-HTP was also found to be a treatment of choice in the prophylaxis of migraine [159,160]. In subjects who are predisposed to headache, 5-HTP can change the central nervous system (CNS) abnormalities underlying the mechanism of migraine [161]. In another study, two weeks after treatment with 5-HTP, a significant decrease in the number of days with headache was observed [162]. 

Ataxia is a clinical manifestation indicating dysfunction of the parts of the nervous system that coordinate movement, such as the cerebellum. Some features of cerebellar ataxia have been reported to regress partially with long-term administration of 5-HTP [163], with a significant decrease in the kinetic score, indicating an improvement in coordination, although the effect may sometimes be partial [164]. 

Lower levels of 5-HTP were found in women with fibromyalgia (FM) in comparison with controls, indicating that the dysregulation of the catecholamine and indolamine pathway in patients with FM may contribute to the physiopathology of this syndrome [165,166]. The treatment with 5-HTP significantly improved all the clinical parameters studied in 50 patients with primary FM syndrome, with only mild and transient side effects reported [167]. 

Concentrations of 5-HTP are lower in dementia of the Alzheimer’s type (DAT) cerebrospinal fluid (CSF) than in a corresponding fraction of control CSF. Therefore, the serotoninergic system is involved in DAT and could be considered for a diagnostic test for DAT [168]. 

5-HTP has a long history of use as a therapy of Parkinson’s disease (PD) [169,170]. In Parkinsonian patients, no effects on gastrointestinal absorption of 5-HTP were observed with co-administration of L-dopa with 5-HTP and decarboxylase inhibitors [171]. Moreover, in a single-center, randomized, double-blind, placebo-controlled, cross-over trial, patients receiving placebo and 50 mg of 5-HTP daily over a period of 4 weeks experienced a significant improvement of depressive symptoms during treatment compared with placebo, providing preliminary evidence of the clinical benefit of 5-HTP for treating depressive symptoms in PD [172].

#### 8.2.5. Effect of 5-HTP on Myoclonus

5-HTP is useful in the treatment of patients with posthypoxic intention myoclonus [173], palatal myoclonus [174], and cherry red spot-myoclonus syndrome [175]. The administration of 5-HTP and carbidopa dramatically improved the action myoclonus and reduced the amplitude of giant somatosensory evoked potentials [176], whereas the combination of sodium valproate and 5-HTP was useful to control a spinal segmental myoclonus characterized by symmetric, rhythmic contractions of the abdomen [177].

#### 8.2.6. Effect of 5-HTP on Obesity

The effect of 5-HTP on feeding behavior, mood state, and weight loss was studied. 5-HTP promoted decreased food intake and weight loss as well as typical anorexia-related symptoms without changes in mood state during the period of observation [178], with a consistent presence of early satiety and a consequent reduction in carbohydrate intake [179]. Moreover, treatment with 5-HTP prompted a decrease in BMI due to an increased feeling of satiety [10].

#### 8.2.7. Effect of 5-HTP on Prolactin

Oral administration of 5-HTP significantly increases plasma human prolactin, suggesting that the serotonergic mechanism is involved in the regulation of prolactin secretion in humans [180]. The maximum downregulation of prolactin release occurs when 5-HTP is administered in 4 h intervals [181] and subacute serotonergic stimulation with oral 5-HTP with the peripheral decarboxylase inhibitor carbidopa resulted in prolactin but not aldosterone release [182].

#### 8.2.8. Antioxidant, Anti-inflammatory, and Analgesic Effects of 5-HTP

5-HTP also exerts radical scavenging activities. 5-HTP showed higher hydroxyl radical scavenging effects when compared to vitamin C [183] and was also effective on hyperglycemia-induced oxidative stress [184]. Moreover, 5-HTP was found to preserve membrane fluidity in the presence of oxidative stress [185]. 5-HTP significantly reduced tert-butylhydroperoxide-induced oxidative damage in human fibroblast cells and protected these cells against oxidative DNA damage [186]. 5-HTP was also found to inhibit the lipopolysaccharide (LPS)-induced expression of NO and interleukine-6 (IL-6), playing a role in extracellular signal-regulated protein kinase (ERK) activation, cyclooxygenase-2 (COX-2,) and LPS-induced inducible nitric oxide synthase (iNOS). By acting as a reactive oxygen species (ROS) scavenger, 5-HTP has the potential for use in the treatment of inflammatory diseases and as an analgesic [187].

Figure 11 summarizes some effects of 5-HTP on humans.

## 9. Toxicology of 5-HTP

An excess of 5-HTP may be responsible for serotonin syndrome (see Section 8.2.1) and an excessive treatment was found to be associated with severe side effects, including behavioral disturbances, abnormal mental functions, and intolerance. Clinically relevant ingestion of 5-HTP in dogs was found to result in a potentially life-threatening syndrome resembling serotonin syndrome in humans, which requires prompt and aggressive care [188]. After 5-HTP administration, the endogenous serotonin levels increased by fourfold in the rat plasma and brain, associated with profound hyperthermia, oxidative stress, and NO upregulation [106].

Toxicity issues have also been raised in 5-HTP. The use of L-tryptophan, the precursor of 5-HTP, as a dietary supplement was suspended in 1989 due to the occurrence of eosinophilia–myalgia syndrome (EMS), a rare, sometimes fatal neurological condition including debilitating myalgia and marked peripheral eosinophilia, that was traced to contaminated synthetic tryptophan from a single manufacturer (Showa Denko) [189,190]. Therefore, 5-HTP has been under vigilance by consumers, industry, academia, and government for its safety. With the possible exception of one unresolved case of a Canadian woman, no definitive cases of toxicity have emerged despite the worldwide usage of 5-HTP for the last 20 years. Neither toxic contaminants similar to those associated with L-tryptophan, nor the presence of any other significant impurities have been detected in several sources of 5-HTP. Speculations concerning the chemistry and toxicity of infinitesimal concentrations of a minor chromatographic peak (peak X) found in some 5-HTP samples lack credibility, due to possible chromatographic artifacts [5]. Further studies found no significant evidence of EMS in rats receiving high-dose 5-HTP for 1 year [191]. Based on accurate mass, tandem mass spectrometric analysis, and comparison with authentic standard compound analysis, peak X was determined to be 4,5-tryptophan-dione, a putative neurotoxin. Because 4,5-tryptophan-dione was found in case-implicated 5-HTP as well as six over the counter samples, some cause for concern in terms of the safety of such commercial preparations of 5-HTP was raised [7]. Indeed, it was demonstrated that samples of commercially available 5-HTP analyzed by HPLC-MS contained three or more contaminants of the peak X family [8]. Two other 5-HTP contaminants, peak E (1,1′-ethylidenebis(L-tryptophan)) and peak-UV5 (3-anilinoalanine), were found to contribute to the pathogenesis of EMS, or may be surrogates for other chemicals that induce EMS [192,193,194]. Another contaminant of tryptophan, peak AAA, was defined as a statistical contaminant and was identified as two distinct isomers: peak AA(1) as (S)-2-amino-3-(2((S,E)-7-methylnon-1-en-1-yl)-1H-indol-3-yl) propanoic acid and peak AAA(2) as (S)-2-amino-3-(2-((E)-dec-1-en-1-yl)-1H-indol-3-yl) propanoic acid [6]. Interestingly, after replacement with 5-HTP not containing impurities, eosinophilia was resolved [195].

## 10. Conclusions

The use of 5-HTP dates back to the first half of the 20^th^ century, where it was recognized as an important precursor of the neurotransmitter serotonin. Its use increased after the discovery of some toxic impurities in commercial tryptophan, although further analyses also found similar impurities in 5-HTP preparation. The main application of 5-HTP is as a support for serotonin depletion and is considered an interesting alternative to the use of SSRIs. The main natural source of 5-HTP is the seeds of the African plant *Griffonia simplicifolia*, which also produces interesting β-carboline alkaloid derivatives. However, the seasonal and regional variations of this plant limit the output of 5-HTP. For this reason, recent biotechnological approaches have been developed by using recombinant genes and genetic engineering of both bacteria (mainly *E. coli*) and yeasts, for the economically and environmentally sustainable production of 5-HTP. These methods are based on biochemical reactions (such as TPH catalysis) and the use of alternative cofactors to improve the biosynthetic ability of bacterial and fungal cells to produce 5-HTP at industrial levels. While on the one side, the use of 5-HTP may be not recommended in humans (as in the case of serotonin syndrome), on the other, the molecule (drug or dietary supplement) has been proved to be effective to treat neurological and metabolic diseases. The majority of clinical trials are on depression, anxiety, panic attacks, and sleep disorders; however, the molecule has been demonstrated to be a promising support to reduce food/feed intake and to be potentially used for metabolic diseases like obesity and diabetes. A few studies also indicate its potential in neurodegenerative diseases like Alzheimer’s and Parkinson’s disease. Future developments are focused on safer 5-HTP production. The improvement of 5-HTP yield and the removal of toxic impurities by biotechnological transformation will allow a sustainable and safer 5-HTP production for both pharmaceutical and nutraceutical industries.

## Figures and Tables

**Figure 1 ijms-22-00181-f001:**
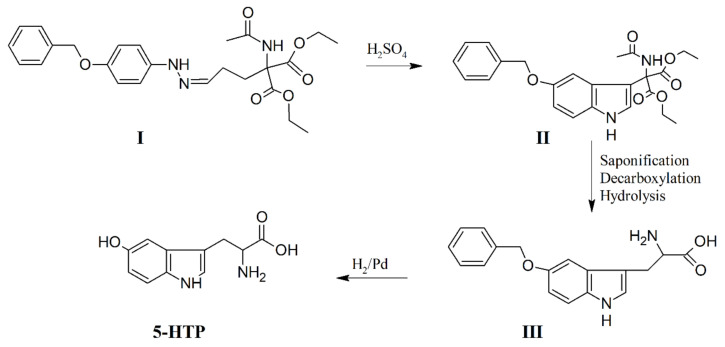
Chemical synthesis of 5-hydroxytryptophan (5-HTP). From [54], modified.

**Figure 2 ijms-22-00181-f002:**
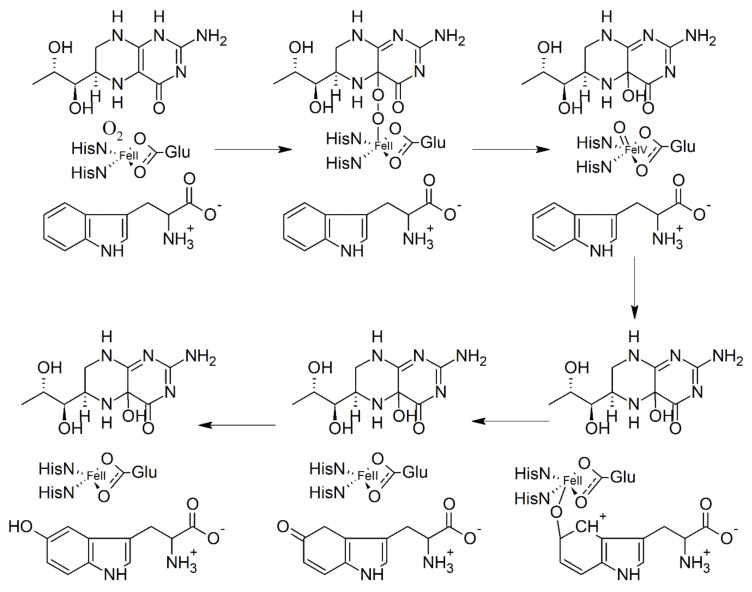
Hydroxylation of tryptophan by tryptophan hydroxylase (TPH). See text for explanation. From [58], modified.

**Figure 3 ijms-22-00181-f003:**
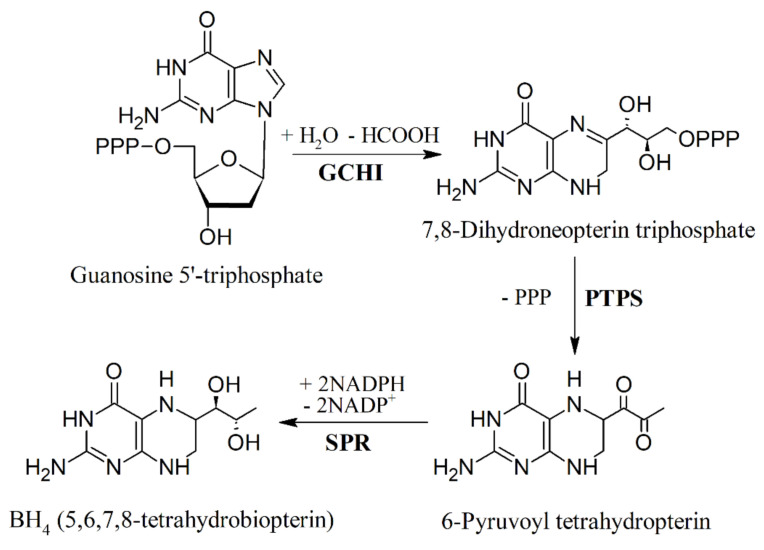
Biosynthesis of BH_4_ in mammals. Modified from [61].

**Figure 4 ijms-22-00181-f004:**
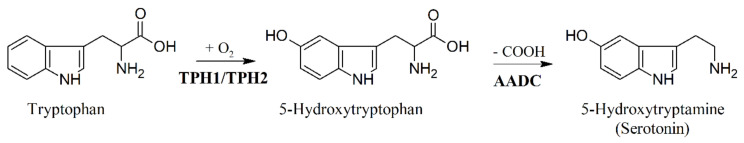
Biosynthetic pathway of serotonin (5-HT). Tryptophan is hydroxylated by the two forms of TPH to yield 5-HTP, which is then decarboxylated by the aromatic amino acid decarboxylase (AADC) to serotonin.

**Figure 5 ijms-22-00181-f005:**
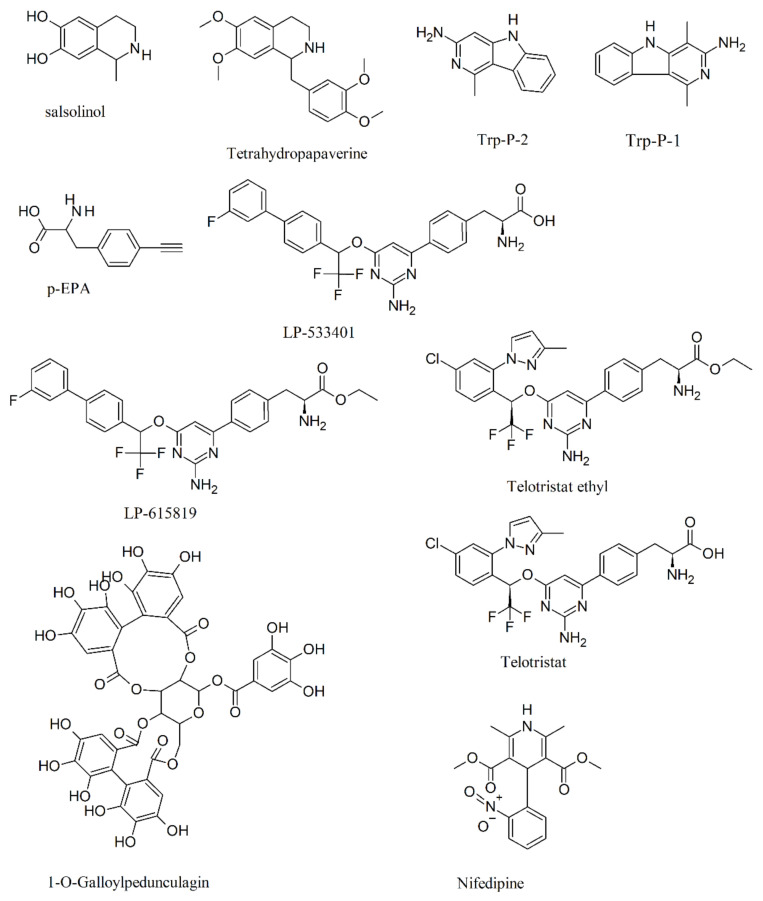
Structure formulae of some TPH inhibitors.

**Figure 6 ijms-22-00181-f006:**
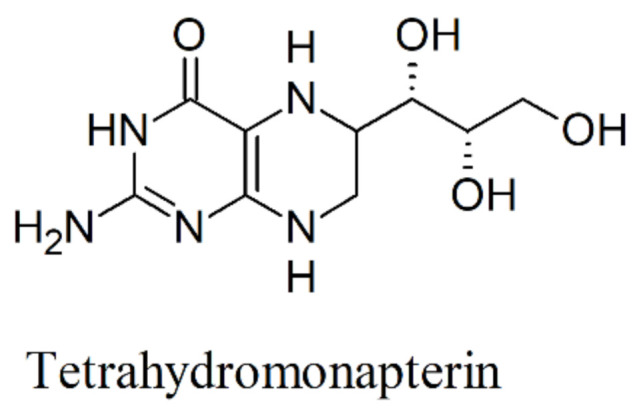
Chemical formula of tetrahydromonapterin.

**Figure 7 ijms-22-00181-f007:**
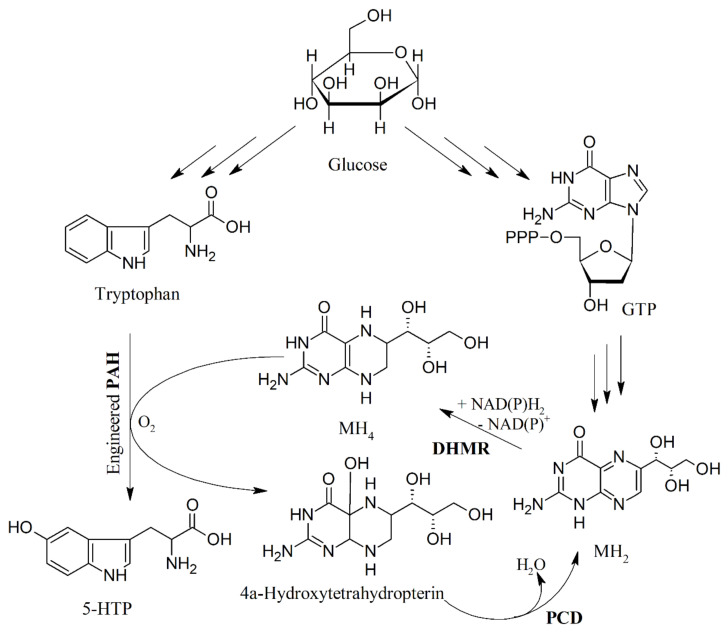
Production of 5-HTP from tryptophan in *E. coli* by engineered PAH and the utilization of tetrahydromonapterin (MH_4_) instead of BH_4_. DHMR, dihydromonapterin reductase; PAH, phenylalanine 4-hydroxylase; PCD, pterin-4α-carbinolamine dehydratase; MH_2_, dihydromonapterin. Modified from [88].

**Figure 8 ijms-22-00181-f008:**
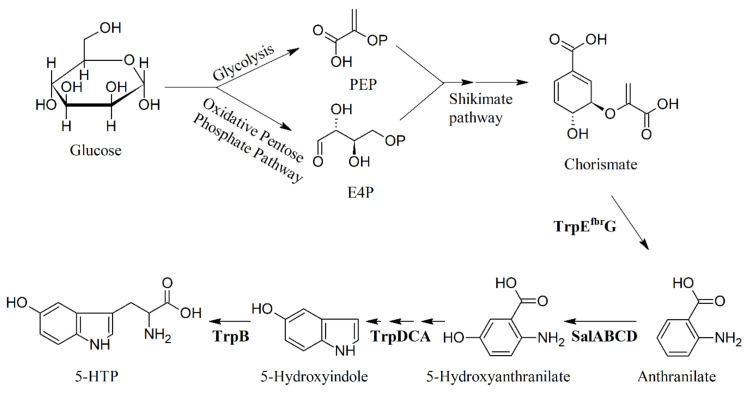
Production of 5-HTP from glucose in engineered *E. coli* by using a novel salicylate 5-hydroxylase. PEP, phosphoenol pyruvate; E4P, erythrose-4-phosphate; TrpE^fbr^G, anthranilate synthase (from a feedback resistance mutant); SalABCD, salicylate 5-hydroxylase; TrpDCA and TrpB, *E. coli* native tryptophan biosynthetic enzymes. Adapted from [90].

**Figure 9 ijms-22-00181-f009:**
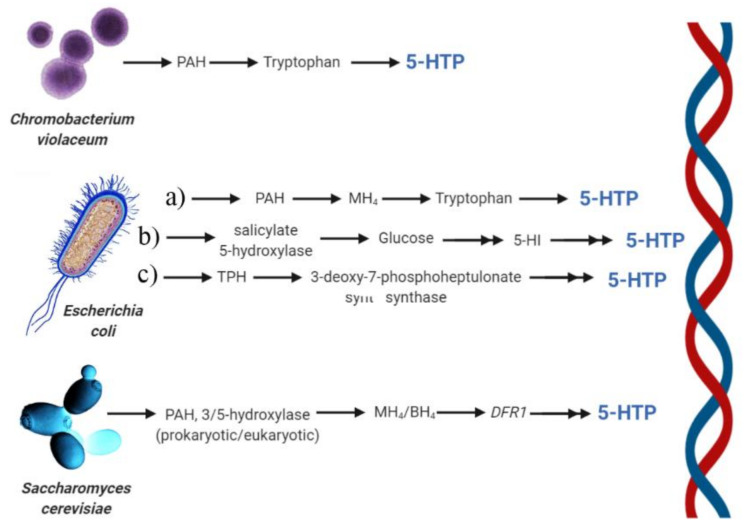
Summary of heterologous production of 5-HTP. *Chromobacterium violaceum*, with the first use of PAH for tryptophan hydroxylation to 5-HTP. *Escherichia coli*, with a) utilization of MH_4_ instead of BH_4_ as the native pterin coenzyme for 5-HTP synthesis, b) the biosynthesis of 5-HTP with the use of a novel salicylate 5-hydroxylase that uses glucose as the substrate for the production of the non-natural substrate anthranilate to 5-hydroxyanthranilate (5-HI), c) substitution of the promoter of *aroHfbr* gene encoding 3-deoxy-7-phosphoheptulonate synthase to feed the biosynthetic pathway of tryptophan production. The use of yeasts with the heterologous expression of either a prokaryotic PAH or eukaryotic tryptophan 3/5-hydroxylase, together with enhanced synthesis of the two cofactors MH_4_ or BH_4_; the native *Saccharomyces cerevisiae* gene, *DFR1*, which encodes dihydrofolate reductase to catalyze tetrahydrofolate, plays a pivotal role in 5-HTP synthesis by regenerating MH_4_. Created with BioRender.com.

**Figure 10 ijms-22-00181-f010:**
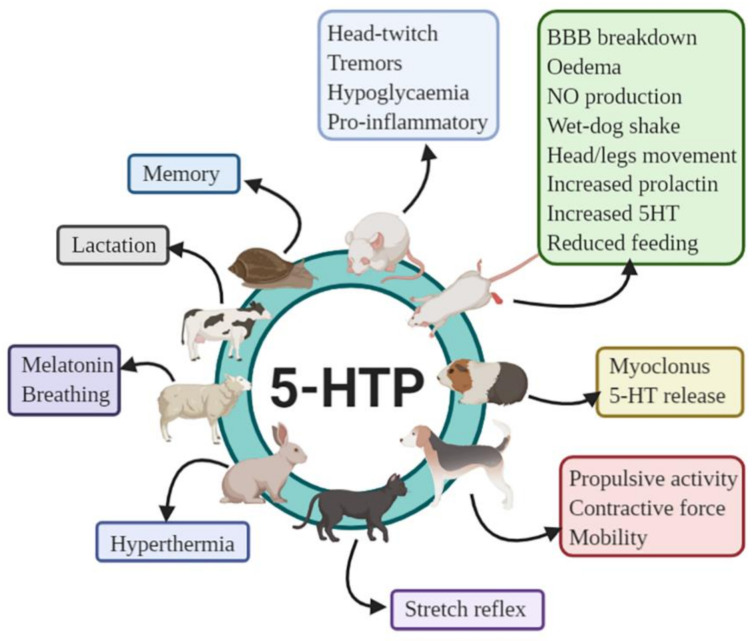
Summary of effects of 5-HTP on some animals. In terrestrial snails, 5-HTP reinstates the context memory, whereas in mice, 5-HTP produces a characteristic head twitch and tremors, induces hypoglycemia, and suppresses inflammation by inducing pro-inflammatory mediators. In rats, most of the listed effects of 5-HTP are dose dependent. In guinea pigs, 5-HTP enhances myoclonus and facilitates the luminal 5-HT release. In dogs, 5-HTP increases propulsive activity, contractive force, and mobility index. Injection of 5-HTP facilitates the stretch reflex in cats and produces a fall in temperature in rabbits. In sheep, 5-HTP increases serum melatonin and foetal breathing movements and stimulates an autocrine–paracrine adaptation to lactation in cows. Created with BioRender.com.

**Figure 11 ijms-22-00181-f011:**
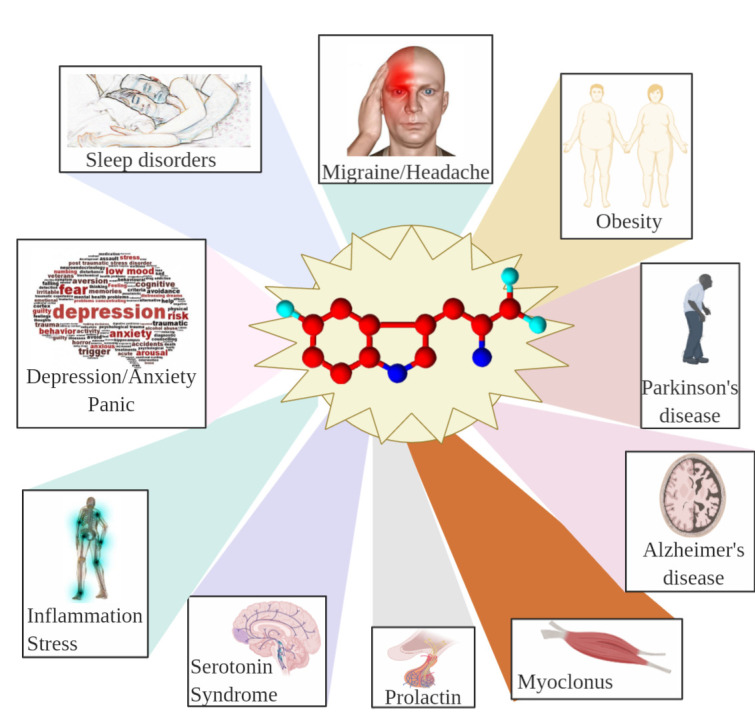
Summary of some effects of 5-HTP on humans. 5-HTP is a treatment of choice in the prophylaxis of migraine and headache and promotes decreased food intake and weight loss in obese patients. 5-HTP is used for treating depressive symptoms in Parkinson’s disease and may be used as a diagnostic test for Alzheimer’s disease. 5-HTP is useful to control some forms of myoclonus and significantly increases plasma human prolactin. Excessive 5-HTP generates serotonin syndrome. 5-HTP has the potential for use in the treatment of inflammatory diseases and oxidative stress. As a precursor of 5-HT, 5-HTP treatment is used to reduce depression, anxiety, and panic attacks. 5-HTP is associated with an increase in rapid eye movement (REM) sleep and reduces sleep disorder. Created with BioRender.com.

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
