# Peer review of "5-Hydroxytryptophan (5-HTP): Natural Occurrence, Analysis, Biosynthesis, Biotechnology, Physiology and Toxicology"

_ijms, 2020, doi:10.3390/ijms22010181_

Round 1

Reviewer 1 Report

In general, the author provided a good information on each of the topics covered in this manuscript. However, this manuscript still needs to describe the strategy that was implemented to carry out this review, which is essential for a review since it allows to know the search limits in years, the inclusion and exclusion criteria, the number of articles that entered into this study, which are the databases that were used, among others. Authors are recommended to complement these information.

Author Response

I thank very much the reviewer for this important comment. A new paragraph (1. Databases, exclusion and inclusion criteria) now describes the missing information.

Reviewer 2 Report

The work entitled “5-Hydroxytryptophan (5-HTP): natural occurrence, analysis, biosynthesis, biotechnology, physiology and toxicology” represents a valuable study on 5-HTP nature. In the reviewer opinion it requires minor corrections before further processing. Please find some comments that should be considered in the revision of the manuscript.

  1. The manuscript describes well the effects associated with 5-HTP. However, information on 5-HT is lacking. Deficiency in 5-HT is described as one of the causes of depression, pulmonary hypertension, and even behavioral defects and gait disturbance. It is worth discussing the genetic background and checking the research done on knockout models.
  2. Are there any known data on 5-HT deficiencies, the basis of these deficiencies, and their frequency in the population? If yes please add them to manuscript.
  3. Figure 9 and 10 – I suggest to reduce the size of these figures. Please specify what kind of effects are presented on Figures (both figures legends and graphics). They are illegible in their present form - What is the specific impact - decrease, increase, weakness, etc. Also fill in the names - Alzheimer's disease, Parkinson's disease
  4. I would be happy to see a corresponding figure depicting heterologous 5-HTP production

Minor: Page 10, line 321: What kind of acid?

Author Response

Q: The manuscript describes well the effects associated with 5-HTP. However, information on 5-HT is lacking. Deficiency in 5-HT is described as one of the causes of depression, pulmonary hypertension, and even behavioral defects and gait disturbance. It is worth discussing the genetic background and checking the research done on knockout models.

A: Despite the importance of 5-HT, this molecule has been described only as part of the biochemical pathway that involves 5-HTP and as a consequence of 5-HTP availability. By considering that much work has been done and reviewed on 5-HT, I preferred to focus the attention of the reader mainly on 5-HTP.

Q: Are there any known data on 5-HT deficiencies, the basis of these deficiencies, and their frequency in the population? If yes please add them to manuscript.

R: Please see the comment above.

Q: Figure 9 and 10 – I suggest to reduce the size of these figures. Please specify what kind of effects are presented on Figures (both figures legends and graphics). They are illegible in their present form - What is the specific impact - decrease, increase, weakness, etc. Also fill in the names - Alzheimer's disease, Parkinson's disease

R: Both figures have now extended legends that explain the effects. The term disease has been added. The dimension of figures has been reduced.

I would be happy to see a corresponding figure depicting heterologous 5-HTP production

R: It was a pleasure to add a figure depicting heterologous 5-HTP production

Minor: Page 10, line 321: What kind of acid?

R: I inserted gastric hydrochloric before acid. Thanks for noticing this.